# Protocol for the evaluation of a pilot implementation of essential interventions for the prevention of cardiovascular diseases in primary healthcare in the Republic of Moldova

Dylan Collins,[1] Angela Ciobanu,[2] Tiina Laatikainen,[3] Ghenadie Curocichin,[2] Virginia Salaru,[2] Tatiana Zatic,[2] Angela Anisei,[2] Jill Farrington[4]

¹University of British Columbia, Vancouver, British Columbia, Canada
²World Health Organization, Chisinau, The Republic of Moldova
³Epidemiology and Health Promotion, National Institute for Health and Welfare, Helsinki, Finland
⁴World Health Organization, Copenhagen, Denmark

**Correspondence to**
Dr Dylan Collins;
dylan.collins@alumni.ubc.ca

## ABSTRACT

**Introduction** Nearly 90% of all deaths in the Republic of Moldova are caused by non-communicable diseases, the majority of which (55%) are caused by cardiovascular diseases (CVD). In addition to reducing premature mortality from CVD, it is estimated that strengthening primary healthcare could cut the number of hypertension-related hospital admissions and diabetes-related hospitalisations in half. The aim of this evaluation is to determine the feasibility of implementing and evaluating essential interventions for the prevention of CVD in primary healthcare in the Republic of Moldova, with a view towards national scale-up.

**Methods and analysis** A national steering group including international experts will be convened to adapt WHO Package of Essential NCD Intervention from Primary Healthcare in Low Resource Settings protocols 1 and 2 to the health system of the Republic of Moldova, develop and conduct training of primary healthcare workers and test a core set of indicators to monitor the quality of care and change in clinical practice. To evaluate the impact of this pilot implementation, a pragmatic, sequential mixed methods explanatory design, composed of quantitative and qualitative strands of equal weight, will be used. Twenty primary healthcare centres will be selected and randomised to the training and implementation arm (n=10) and the usual care arm (n=10). At baseline and 12 months follow-up, a standardised data collection form will be piloted to extract data directly from patient paper records in order to estimate the change in clinical practice. Semi-structured interviews and interclinic peer workshops will be conducted at 12 months follow-up, and qualitative data collected from these formats will be analysed thematically for explanatory themes that relate to the quantitative findings.

**Ethics and dissemination** Ethical review and approval has been obtained. Findings of the evaluation will be shared in a project report to key stakeholders, presented back to participants and written into a manuscript for an open access peer-reviewed scientific journal.

## Strengths and limitations of this study

► To our knowledge, this is the first description of adapting and piloting WHO essential non-communicable disease interventions in primary healthcare in a low-income or middle-income country and provides a methodological example to other jurisdictions.

► A mixed methods design allows for a greater understanding of the potential barriers and facilitators to implementation and can inform future health systems development.

► Primary healthcare facilities will be selected from different regions of the Republic of Moldova in order to pilot implementation in a variety of contexts throughout the country.

► Since this is an evaluation of a pilot implementation, the sample size is based on pragmatism and not statistical power.

► We are unable to include patient perspectives and experience in the evaluation, which is an important aspect of healthcare quality.

## INTRODUCTION

Globally, non-communicable diseases (NCDs) account for more than one-half of the global burden of disease.[1] In 2016, an estimated 41 million deaths were due to NCDs, of which nearly half were due to cardiovascular diseases (CVD).[2] Primary healthcare systems play an important role in the prevention, early detection and appropriate management of these diseases, but many nations lack primary healthcare capacity.[3 4]

To support national governments to realise their commitments in reducing the burden of NCDs, as agreed in the United Nations Political Declaration on NCDs, the World Health Assembly endorsed the WHO Global Action Plan for the Prevention and Control of NCDs 2013–2020. To support implementation of

this Action Plan, WHO has identified a set of cost-effective policy options ('best buys') for the prevention and control of NCDs within countries.[5]

The Republic of Moldova (henceforth 'MDA') is located in Eastern Europe, between Ukraine and Romania; the Capital and largest city is Chisinau. By gross domestic product per capita, MDA is one of the poorest countries in the WHO European Region and it is estimated that 21.9% of citizens live below the absolute poverty line of US$1 per day.[6]

### Non-communicable diseases are a leading cause of death in MDA

While NCDs are a global epidemic, MDA ranks among the countries most affected. Nearly 90% of all deaths in MDA are caused by NCDs, the majority of which (55%) are caused by CVD.[7] In 2016, the probability of dying prematurely from any of the four major NCDs (CVDs, cancer, diabetes, chronic respiratory disease) was 24.9%; almost twice as high for men (33.7%) as women (17.3%).[8] Men and people residing in rural areas are disproportionally impacted by CVD and represent key populations for public health intervention.[7]

This burden is driven by some of the highest rates of NCD risk factors, including tobacco and alcohol use, in the WHO European region indicated by a 2013 Stepwise Approach to Surveillance (STEPS) survey.[9] One in four (25.3%) Moldovans smoke tobacco and this rate nearly doubles in men.[9] Among adults aged 18–69 years, 61.9% currently consume alcohol and one in five people have engaged in heavy episodic drinking (six or more drinks on any one occasion in the past 30 days).[9]

The overall prevalence of obesity among adults is 22.9%, being higher among women (28.5%) as compared with men (17.8%).[9] The prevalence of raised blood pressure (defined as systolic blood pressure ≥140 mm Hg and/or diastolic blood pressure ≥90 mm Hg or currently taking medication for raised blood pressure) among MDA's adult population is 39.8%, and 76.2% of these patients are not on blood pressure-lowering medication.[9] A total of 12.3% of the population have a blood glucose level of ≥6.1 mmol/L, and 29.4% of the population has a total blood cholesterol level of ≥5 mmol/L.[9] It is estimated that one in five (23.0%) people aged 40–69 years have a 10-year fatal or non-fatal CVD risk of over 30% (including those with an existing CVD).[9]

### Primary healthcare in MDA and commitment to NCDs

According to the Constitution of MDA of 1994, citizens are entitled to a free of charge minimum package of essential health services, including primary healthcare. However, resource constraints have made it difficult to offer these services and significant gaps in care exist.[10] According to the most recent data (2010), there were 5.3 family doctors per 10 000 inhabitants and 25.9 specialist doctors per 10 000 inhabitants. In rural areas these rates are halved, leading to human resource shortages in primary care.[10] Approximately 17% of practising physicians in MDA work

in primary healthcare, and 92% of them rely on paper-based clinical records.[6] The most recent estimate (2009) states that there are approximately 630 primary healthcare centres throughout the country, or 21.2 centres per 100 000 people.[6]

Despite these health system challenges, the Government of MDA is committed to improving primary healthcare capacity for NCDs. It is estimated that 60% of hypertension-related hospital admissions (about 12 000 annually) and 40% of diabetes-related hospitalisations (about 5000 annually) could be prevented through strengthened primary healthcare for these conditions, including better identification and management of those at increased CVD risk.[11]

Given the need and international policy support for addressing this gap in NCD care, there was a favourable window of opportunity to act with impact. As such, strengthening primary healthcare was set out as one of the main commitments in the Action Programme of the Government of MDA 2016–2018.[12] To do this requires the development of simplified clinical protocols, in-person training programmes for nurses and doctors and a core set of indicators to monitor and evaluate changes in the quality of care.

### Essential interventions to prevent cardiovascular diseases in primary healthcare

In order to build capacity in primary healthcare and ultimately prevent premature mortality from CVD in MDA, a study was envisioned to adapt and pilot the WHO Package of Essential NCD Intervention from Primary Healthcare in Low Resource Settings (WHO PEN).[3] WHO PEN includes simplified clinical protocols which together cover the integrated management of hypertension and diabetes, as well as education and counselling on healthy behaviours aimed to prevent CVD. The central strategy of this integrated approach is the use of total cardiovascular risk assessment to stratify and target individuals at high CVD risk, a process considered to be a 'best buy' intervention by WHO.[5]

These interventions are aimed at tackling areas identified in a 2014 WHO assessment of challenges and opportunities for better NCD outcomes in MDA.[13] This includes shortcomings among health workers in the identification and management of individuals with increased cardiovascular risk. The interventions are expected to add to the current quality of care by targeting interventions (non-pharmacological and/or pharmacological) to those at highest risk who stand to gain the most in absolute cardiovascular risk reduction, while also emphasising improvements in the organisation of care. The intervention also includes practical face-to-face training and follow-up implementation support. Current practice underuses these medical strategies and guidelines (eg, CVD risk score directed primary prevention), in addition to limited task sharing with non-physician health workers (eg, nurses) in these care pathways.[13] At the study's inception, there were no known developments beyond the

scope of this project that could change clinical practice for NCDs in primary healthcare.

Since the use of WHO PEN was unprecedented in MDA, the Ministry of Health, Labour and Social Protection convened a national steering group to lead the adaptation and pilot process, with the goal of using the findings for future health systems development. Led by the primary healthcare division of the Ministry of Health, the steering group comprises representatives from the Nicolae Testemitanu State University of Medicine and Pharmacy and the National Public Health Agency. The national steering group is supported by an international team of experts coordinated jointly by the WHO Regional Office for Europe and WHO Country Office in MDA.

## AIM AND OBJECTIVES
### Aim
The aim of the evaluation is to determine the feasibility of implementing and evaluating essential interventions for the prevention of cardiovascular disease in primary healthcare in MDA, with a view towards national scale-up.

### Objectives
#### Primary objectives
1. Assess the ability to implement MDA-adapted WHO PEN protocols 1 and 2 in pilot primary healthcare centres.
2. Determine the feasibility of collecting quantitative data required for future studies of effectiveness from the existing informal paper clinical record system.

#### Secondary objectives
1. Determine the baseline performance of primary healthcare services with respect to essential interventions for the prevention and management of CVD.
2. Estimate the change in performance of pilot primary healthcare centres after 12 months of protocol implementation and compare this with control clinics using usual care.

## METHODS AND ANALYSIS
### Overview of process and design
An overview of the methods used to adapt, pilot and evaluate essential interventions for CVD in primary healthcare in MDA are summarised by the following seven steps, which are planned to occur from September 2016 to May 2019.

### Step 1: adaptation of WHO PEN protocols to the national context
Under the direction of the national steering group, WHO PEN protocols 1 and 2 will be compared and contrasted to national disease-specific guidelines. The WHO PEN protocols will then be adapted to ensure consistency with the organisation, culture and availability of resources of the health system, while ensuring that they remain simple clinical decision support tools.

### Step 2: development of a training package for primary healthcare workers
A 3-day training package will be developed under the direction of the national steering group in order to provide in-person theoretical and practical training to nurses and doctors working in primary healthcare. This will include lectures, clinical case studies and practical exercises that embrace the experience and knowledge of participants.

### Step 3: collection of baseline data
According to the Ministry of Health process, a list of 20 primary healthcare clinics will be nominated and provided to the working group. They will then be randomised into an intervention group arm (n=10) and control arm (n=10). Data for quantitative indicators will be extracted from all 20 clinics by randomly sampling individual paper-based patient records from all primary healthcare units using a standardised data collection instrument. This will be done before randomisation by a specially trained group of postgraduate medical trainees, such that neither the clinics nor the data extractors will know the allocation of each clinic to intervention or control arm.

### Step 4: training staff in pilot clinics
All doctors and nurses from the primary healthcare centres in the intervention arm will be invited to be trained together by a national team of experts in groups of approximately 30. It is estimated that up to 200 health workers will be trained in total. At the end of training, each primary healthcare team will pass through evaluation at the University Centre for Simulation in Medical Training using objective structured clinical exams and get feedback from trainers and peers.

### Step 5: implementation of protocols
Trained participants from the 10 primary healthcare clinics in the intervention arm will then be free to implement the clinical protocols and change their clinical practice, without incentives, for 12 months. During this time, a team of national experts will be created to offer support (distance and on-the-job) to the primary healthcare centres in the intervention arm. All 10 clinics in the intervention arm will receive at least one in-person follow-up support visit.

### Step 6: collection of follow-up data
After 12 months, using the same method and data collection instruments used to collect baseline quantitative data (step 3), data will again be extracted from randomly selected individual paper-based patient records from all 20 healthcare centres. Five primary healthcare centres from the intervention arm will be selected by the national steering group for one-on-one semi-structured interviews with health staff. This will be supplemented by inviting a selection of staff from all 10 health centres in the intervention arm to participate in focus groups. Together, these qualitative data will be analysed thematically for explanatory themes.

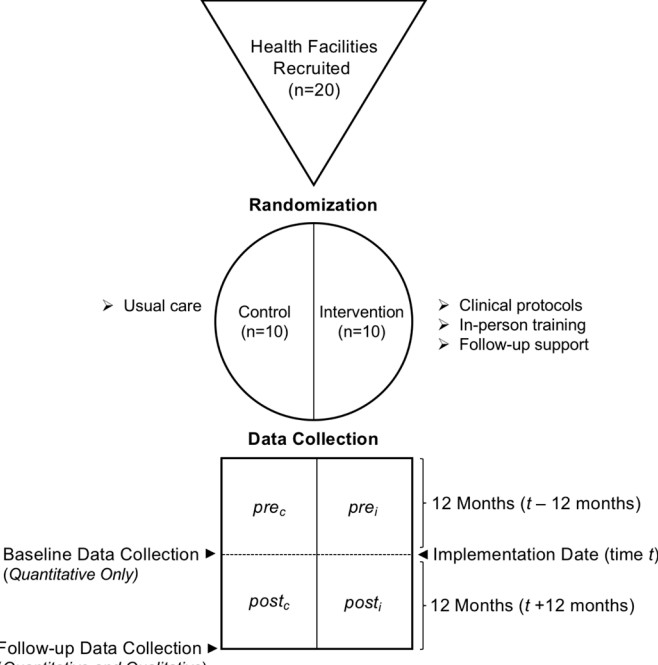

**Figure 1** Illustration using the GATE frame structure[15] of the mixed methods evaluation design. GATE, Graphic Appraisal Tool for Epidemiological studies.

### Step 7: evaluation of results and sharing experience

The findings of the quantitative and qualitative analyses will be integrated in a final report and shared with key stakeholders, including health staff from the participating primary healthcare centres. The results will also be shared at a national conference and in an open-access peer-reviewed journal, in order to inform the future development of primary healthcare capacity in MDA.

### Methodological design

A pragmatic, sequential mixed methods explanatory design, composed of quantitative and qualitative strands of equal weight, will be used (figure 1). This design was chosen because it allows for the use of qualitative data to enlighten and explain the quantitative findings, including but not limited to the feasibility of collecting data from paper-based records, the contextual factors affecting guideline implementation, changes in clinical practice and optimisation for the future.

A sample size of 20 primary healthcare centres was chosen because it was seen as a good balance of allowing for variation in clinic geography and demography, while still remaining feasible for the pilot implementation. Half of the centres (n=10) will be randomly allocated to the intervention arm and half (n=10) to the control arm. Baseline data will be collected from both intervention and control clinics, ensuring that baseline data are collected before implementation occurs.

Within clinic comparisons will be used to compare the 12 months before randomisation with the 12 months of implementation. Between clinics comparison will be used

to compare the intervention clinics with control clinics during the same time period.

### Eligibility criteria for primary healthcare centres

Health facilities will be nominated by the Ministry of Health for participation based on the following eligibility criteria[1]: primary healthcare facilities must be operating in the public sector as legal entities[2]; primary healthcare facilities must be sampled in a way such that they are geographically distributed evenly across the country; equally from the Central, North and Southern regions of MDAand[3] health facilities must be primary healthcare centres that are managed by family doctors with no specialist doctors working in the facility. These criteria were chosen in order to select a group of clinics that sufficiently reflect the majority of primary healthcare facilities in MDA.

### Randomisation

The clinics will be stratified based on the ratio of patients to family doctors to minimise possible confounding by doctor caseload, and then randomised electronically into two groups of 10 primary healthcare centres.

### Comparison

The 10 primary healthcare centres in the intervention arm will be compared with the 10 primary healthcare centres in the control arm. The control arm will receive no intervention and proceed with usual care.

### Quantitative indicators

Indicators were developed to balance input and process indicators, such as measurement of risk factors and calculation of risk scores, with output (eg, prescribing) and outcome (eg, blood pressure control) indicators. While one of the objectives of this evaluation is to determine the ability to measure these indicators based on routine paper records, we used our existing knowledge of the health system to design indicators which were valuable and likely to be feasible to calculate. Table 1 shows the indicator, the question the indicator seeks to answer and the respective numerator and denominator definitions that will be used in the calculations.

### Data collection and management
#### Quantitative data collection tool

A standardised data collection template has been developed for extracting anonymised patient data from individual paper records (table 2). An online version was also made to allow for data entry on a computer or smartphone. It is estimated to take 15 min to extract data from one patient record since the records are made of blank paper with no formal structure or organisation of health data.

#### Method of randomly sampling patient records

A random sample of the records of patients aged over 18 years, who have visited the medical facility within the past 12 months, will be taken. Since medical records in

**Table 1** Indicators, their numerators and denominators and questions the indicators answer

| Question | Indicator | Numerator | Denominator |
|---|---|---|---|
| Are risk factors being measured? | Proportion of eligible patients who have all risk factor values recorded as required for calculation of risk score. | Patients aged 40 years or older who have visited in the last 12 months who have all measurements required for calculation of risk score within 12 months of the most recent date of visit. | Patients aged 40 years or older who have visited in the last 12 months. |
| Are risk factor measurements being converted to a total risk score? | Proportion of patients aged 40 years or older who have visited in the last 12 months who have all measurements required for calculation of risk score within 12 months of the most recent date of visit, which have a documented risk score. | Patients aged 40 years or older who have visited in the last 12 months who have all measurements required for calculation of risk score within 12 months of the most recent date of visit, which have a documented risk score. | Patients aged 40 years or older who have visited in the last 12 months who have all measurements required for calculation of risk score within 12 months of the most recent date of visit. |
| Are risk scores calculated correctly? | Proportion of patients aged 40 years or older who have visited in the last 12 months who have all measurements required for calculation of risk score within 12 months of the most recent date of visit, which have a documented risk score that is correct. | Patients aged 40 or older who have visited in the last 12 months who have all measurements required for calculation of risk score within 12 months of the most recent date of visit, which have a documented risk score that is correct. | Patients aged 40 or older who have visited in the last 12 months who have all measurements required for calculation of risk score within 12 months of the most recent date of visit, which have a documented risk score. |
| Are patients being risk scored? | Proportion of eligible patients with a documented risk score. | Patients aged 40 years or older who have visited in the last 12 months with a documented risk score. | Patients aged 40 years or older who have visited in the last 12 months with a documented risk score. |
| Are risk scores calculated correctly? | Proportion of eligible patients with a documented risk score that is correct. | Patients aged 40 years or older who have visited in the last 12 months with a documented risk score that is correct. | Patients aged 40 years or older who have visited in the last 12 months with a documented risk score. |
| Are statins prescribed to the correct patients? | Proportion of eligible patients prescribed a statin. | Patients with existing CVD, patients with diabetes aged 40 years or older with high LDL values (as defined based on total CVD risk of SCORE 10%–14% in LDL≥2.6mmol/L; with very high risk SCORE≥15%in LDL≥1.8mmol/L), or patients with a SCORE of ≤9% and LDL≥2.6or total cholesterol≥7.2, or patients with a SCORE of 10%–14% and LDL≥1.8or total cholesterol≥7.2mmol/L, or patients with a SCORE of ≥15%, prescribed a statin. | Patients with existing CVD, patients with diabetes aged 40 years or older with high LDL values (as defined based on total CVD risk of SCORE 10%–14% in LDL≥2.6mmol/L; with very high risk SCORE≥15%in LDL≥1.8mmol/L), or patients with a SCORE of ≤9% and LDL≥2.6or total cholesterol≥7.2, or patients with a SCORE of 10%–14% and a LDL≥1.8or total cholesterol≥7.2mmol/L, or patients with a SCORE of ≥15%. |
| Are statins prescribed correctly based on documented risk score? | Proportion of patients eligible based on documented risk score prescribed a statin. | Patients with a documented risk score as very high risk SCORE≥15% prescribed a statin. | Patients with a documented risk score as very high risk SCORE≥15%. |

**Table 1** Continued

| Question | Indicator | Numerator | Denominator |
|---|---|---|---|
| Are patients with existing disease, who do not require the calculation of a risk score to prescribe satins, prescribed statins? | Proportion of patients with existing CVD prescribed a statin. | Patients with existing CVD prescribed a statin. | Patients with existing CVD. |
| Is the blood pressure of high-risk patients controlled? | Proportion of high-risk patients (SCORE≥15% or DM and age over 40 years) whose last two recorded blood pressure measurements were <130/80 mm Hg. | Patients with a true risk score indicating a very high risk (SCORE≥15%) or DM and age over 40 years whose last two documented blood pressure readings were <130/80 mm Hg. | Patients with a true risk score indicating a very high risk (SCORE≥15%) or DM and age over 40 years. |
| Is the blood pressure of low-risk patients controlled? | Proportion of low-risk patients (SCORE<15%) whose last two recorded blood pressure measurements were <140/90 mm Hg. | Patients with a true risk score indicating <15% whose last two documented blood pressure readings were <140/90 mm Hg. | Patients with a true risk score indicating <15%. |
| Are patients with existing CVD prescribed basic medications to reduce risk? | Proportion of patients with existing CVD prescribed a statin and aspirin and blood pressure-lowering treatment. | Patients with existing CVD prescribed a statin and aspirin and blood pressure-lowering treatment. | Patients with existing CVD. |
| Is the blood glucose of patients with diabetes controlled? | Proportion of patients with diabetes with glycaemic control as defined by last two HbA1c measurements. | Patients with type 2 diabetes whose last two HbA1c measurements were below personal target as defined by MDA-adapted WHO PEN protocol 1. | Patients with type 2 diabetes. |
| Is the blood pressure of patients with hypertension controlled? | Proportion of patients with confirmed hypertension whose systolic blood pressure is <140/90 mm Hg at last two visits. | Patients with confirmed hypertension whose last two blood pressure readings were <140/90 mm Hg. | Patients with confirmed hypertension. |
| What is the prevalence of high blood pressure? | Proportion of people whose last two systolic blood pressure reading are 140 mm Hg or above. | Patients whose last two systolic blood pressure readings were ≥140 mm Hg. | All patients aged over 18 years. |

CVD, cardiovascular disease; DM, diabetes mellitus; HbA1c, glycated haemoglobin; LDL, low-density lipoprotein; SCORE, Systematic COronary Risk Evaluation.

**Table 2** Standardised data collection form used to extract data from individual patient records

| Data collection question | Answer |
|---|---|
| What is your name? (Name of person extracting data) | |
| Date of data extraction (MM-DD-YYYY) | |
| Write the clinic name | |
| Is this a duplicate extraction? | |
| If it is a duplicate extraction, enter the number you and your extraction partner have assigned to this file. | |
| Date of birth (MM-DD-YYYY) | |
| Sex (M/F) | |
| Smoking status (Y/M) | |
| Diagnosis of hypertension (Y/N) | |
| Date of hypertension diagnosis (MM-DD-YYYY) | |
| Can you find one or more blood pressure readings? (Y/N) | |
| Most recent systolic blood pressure | |
| Most recent diastolic blood pressure | |
| Date of the most recent blood pressure measurement (MM-DD-YYYY) | |
| Can you find a second most recent blood pressure reading? (Y/N) | |
| Second most recent systolic blood pressure | |
| Second most recent diastolic blood pressure | |
| Date of the second most recent systolic blood pressure (MM-DD-YYYY) | |
| Diagnosis of diabetes (type 1, type 2, no) | |
| Can you find one or more glycated haemoglobin (HbA1c) measurements? (Y/N) | |
| Most recent HbA1c reading (mmol/mol) | |
| Date of the most recent HbA1c measurement? (MM-DD-YYYY) | |
| Can you find another HbA1c measurement? (Y/N) | |
| Second most recent HbA1c reading (mmol/mol, otherwise specify unit) | |
| Date of the second most recent HbA1c reading? (MM-DD-YYYY) | |
| Can you find one or more total cholesterol measurements? (Y/N) | |
| Most recent total cholesterol reading (mmol/L) | |
| Date of the most recent cholesterol reading (MM-DD-YYYY) | |
| Can you find another cholesterol measurement? (Y/N) | |
| Second most recent cholesterol reading (mmol/L) | |
| Date of the second most recent cholesterol reading (MM-DD-YYYY) | |
| Was the patient prescribed a statin? (Y/N) | |
| What was the date of the statin prescription? (MM-DD-YYYY) | |
| What was the drug and dose? | |
| Does the patient have existing cardiovascular disease (CVD)? (Y/N) | |
| State the type of CVD | |
| Has the patient been prescribed acetylsalicylic acid (ASA or aspirin)? (Y/N) | |
| What was the most recent date that ASA was prescribed? (MM-DD-YYYY) | |
| Has the patient been prescribed antihypertensives? (Y/N) | |
| What was the most recent date that antihypertensives were prescribed? (MM-DD-YYYY) | |
| Can you find a documented ESC SCORE risk score? (Y/N) | |
| Enter the most recent documented ESC SCORE risk score (%) | |
| What was the date the risk score was documented? (MM-DD-YYYY) | |

| Table 2 Continued | |
|---|---|
| **Data collection question** | **Answer** |
| Please record any important notes about the data extraction here. Examples include an error you think may have been made, clarification of the units for measurements (eg, mmol/L vs mg/dL). Or notes that you would like for yourself. | |

ESC, European Society of Cardiology.

MDA are organised alphabetically on shelves, we created a randomly generated list of alphanumeric combinations that allowed for the selection of patient charts at random. For example, an alphanumeric code of 'C24' would correspond to the 24th patient chart in the section of last names starting with the letter C.

The list will be followed in the order that it was generated so as to prevent selection bias. The randomly selected chart will then be checked to see if it meets two inclusion criteria[1]: the patient is aged 18 years or older and[2] the patient visited the health centre within the last 12 months. If the record meets these criteria, data will then be extracted. If it does not, it will be returned to the shelf and the next alphanumeric code on the randomly generated list will be used. This process will be repeated in each clinic until a sample size of 1.2% of the patient population in each clinic is sampled. This proportion was chosen pragmatically such that the average sample per primary healthcare centre would equal 100 unique patients.

### Data analysis

The change in indicators from baseline to follow-up will be calculated for intervention clinics and compared with control clinics (table 1). Subgroup analysis by age, gender and other demographic features may be done as deemed appropriate by the national steering committee. All analyses will account for stratified sampling. Since the health centre is the unit of inference for the outcomes (eg, health centre proportion of eligible patients with a documented CVD risk score), use of an intracluster correlation coefficient is not required for analyses of these outcomes. Age-adjusted and gender-adjusted logistic regression models will be used to analyse the differences in predefined indicators between intervention and control clinics and between baseline and follow-up. The differences in means of continuous variables between the intervention and control clinics and baseline and follow-up will be analysed using age-adjusted and gender-adjusted analysis of variance.

### Qualitative data collection
#### Follow-up support visits

Follow-up visits will be made to each intervention clinic at least once during the implementation timeframe (12 months) to provide ad hoc implementation support. These visits will be conducted by members of the national steering group, who will keep field notes about each visit and provide feedback and support to the health centres.

The perspectives gained through follow-up support visits will be used by the national steering group to develop preliminary data collection tools for semi-structured interviews.

### Semi-structured interviews

A maximum variation sample of half of the intervention clinics (n=5) will be chosen, based on the perceived performance of each clinic by the evaluation steering committee. A pragmatic sample of clinic managers (n=1 per clinic), doctors (n=3 per clinic) and nurses (n=3 per clinic) will be interviewed one-on-one, using a semi-structured format. Interviews will proceed until data saturation has been reached to a maximum of 30 interviews. After obtaining written informed consent, interviews will be of 30–60 min in length, audio-recorded and be transcribed verbatim and analysed thematically using framework thematic analysis.[14] The interviews will be conducted by members of the steering group, but the interviewers will be allocated to participants from health centres with whom they did not provide follow-up support visits.

### Focus group workshop

Participants from all 10 implementation clinics will be invited to a workshop to further collect explanatory qualitative data and to critically reflect on the implementation process. Participants will be a mix of doctors, nurses and managers from the intervention clinics.

Participants will be placed into small groups based on their profession, and asked to complete a standardised worksheet. Each group will be under the guidance of a facilitator, and emergent themes from one-one-one interviews will be used as prompts to each group. The worksheet will allow for each group to directly comment, modify or add to the emergent themes, create new themes and organise themes into categories such as barriers and facilitators.

### Integration of quantitative and qualitative strands

The resulting qualitative data will be analysed thematically using the framework approach, and used to help explain the findings of the quantitative strand.[14] Following the sequential mixed methods design, integration of the qualitative findings with quantitative findings will allow for the interpretation of the results in light of each other. This may include post hoc analysis of effectiveness of some of the quantitative outcomes as appropriate, to further add meaning to the integration of qualitative and quantitative strands.

## Patient and public involvement

Neither patients nor the public were involved in the methodological design.

## DISSEMINATION

Quantitative findings will be summarised and presented back to all intervention clinics during follow-up workshops. A comprehensive project report will be written and shared with key stakeholders. A final report of key findings of the evaluation will be written and submitted to an open access peer-reviewed journal and made available to all study participants so they can use the findings to improve their practice. The findings will be used to evaluate the feasibility of a national scale-up of essential NCD interventions in primary healthcare in MDA.

**Contributors** DC, AC, TL, GC, VS, TZ, AA and JF contributed to the methodological design. DC, AC, TL and JF contributed to writing the manuscript.

**Funding** This study is funded jointly by the Swiss Agency for Development and Cooperation (SDC) and WHO Regional Office for Europe.

**Competing interests** None declared.

**Patient consent for publication** Not required.

**Ethics approval** This project was reviewed by the Research Ethics Committee of the Nicolae Testemitanu State University of Medicine and Pharmacy of the Republic of Moldova and granted permission on 31 May 2017.

**Provenance and peer review** Not commissioned; externally peer reviewed.

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
