## [Reviewer comments · BMJ Open]

ARTICLE DETAILS

TITLE (PROVISIONAL)	Protocol for the Evaluation of a Pilot Implementation of Essential Interventions for the Prevention of Cardiovascular Diseases in Primary Health Care in the Republic of Moldova
AUTHORS	Collins, Dylan; Ciobanu, Angela; Laatikainen, Tiina; Curocichin, Ghenadie; Salaru, Virginia; Zatic, Tatiana; Anisei, Angela; Farrington, Jill

VERSION 1 - REVIEW

REVIEWER	Dr. T.N. Bonten Leiden University Medical Center
REVIEW RETURNED	16-Oct-2018

GENERAL COMMENTS	- Statistics are now not clearly described, please add this.- Methodology: randomization and intervention is not blinded. This might not be required, but then authors need to think about possible bias in the control group. For example: do control group practices also have access to protocols? They might also want to 'implement' something in line with the intervention arm (dilution of the effect). Please consider this.
--

REVIEWER	Abholz Department General Practice, Duesseldorf University, Germany
REVIEW RETURNED	18-Nov-2018

GENERAL COMMENTS	I would not publish Research Applications or pure Descriptions of a planned study- and this is one
--

REVIEWER	Martin Gulliford King's College London
REVIEW RETURNED	15-Feb-2019

GENERAL COMMENTS	This protocol describes a service development project focused on primary care interventions for chronic diseases. The intervention comprises an adaptation of the World Health Organization Package of Essential NCD Intervention from Primary Healthcare in Low Resource Settings for use in Moldova. The evaluation
---

	comprises a mixed methods evaluation based on cluster randomisation of 20 primary care centres. The paper describes work that is important in its national context. It might be useful for countries in a similar situation. On the other hand, the work may not be particularly novel as many countries are addressing these issues. The activities comprise audit and quality improvement activities. I did not see dates during which the proposed work was to be conducted. There are many abbreviations that are not initially explained. The basis for randomisation is uncertain. Is it ethically justified to include a control group that will not benefit from these interventions? A step-wedge design might have been better in this context. From a scientific perspective, some aspects of the manuscript appear simplistic. For example, the analysis plan reads 'The change in indicators from baseline to follow-up will be calculated for intervention clinics and compared with control clinics.' Though it is not made clear what statistical procedures will be used nor how cluster randomisation will be accounted for. Overall, this plan describes good work, but it does not make a convincing case that the cause of science requires this protocol to be published.
--	---

REVIEWER	E.P. Moll van Charante Amsterdam University Medical Center, The Netherlands
REVIEW RETURNED	24-Feb-2019

GENERAL COMMENTS	In this article, the authors describe the protocol of a study aimed to determine the feasibility of implementing and evaluating essential interventions for the prevention of CVD in primary health care in Republic of Moldova. Population wide interventions on the prevention of NCDs, such as cardiovascular disease, in low- and middle income countries are both urgently needed and rare, and the authors (and other initiators involved) are to be commended on this initiative. While the mixed methods approach is likely to strengthen the validity of overall study findings, in my view, there are some aspects of the study design that need further clarification or adaptation to improve overall understanding and impact. Firstly, it would be very helpful to add something on how the intervention may feed into the needs and limitations of the current care by GPs and nurses. While there is ample mentioning of highly prevalent CV risk factors in the population, it appears less clear what aspects of professional care are expected to increase their overall effectiveness and why. For instance, is a more proactive outreach to the high-risk population part of the envisioned, improved preventive strategy, and if so, will this also be matched by a strengthened workforce to address the needs encountered? Or does the training of professionals (step 4) focus on presumed shortcomings in the cardiovascular risk management itself? And if so, how is it expected to both add to the current quality of care and related professional needs? This also emphasizes the need to further describe the 'care as usual': what professional care is
---

currently provided and according to which guidelines or protocols is this executed? And are there any current developments that may impact care in the control group during the study period? Secondly, since this study aims to both develop and test/evaluate the intervention and outcome parameters, this may threaten the interpretability and validity of overall study findings. The authors might feel inclined to describe possible solutions to protect the study design from such potential pitfalls.

I have added some specific comments below.

Introduction

Consider adding some information about the current treatment gap (see above) and a (realistic) window of opportunity. Also, the reader might want to understand how the University of British Columbia became involved in this project.

Methods

Study contrast. Next to new developments in the prevention of CVD, Hawthorne effects may reduce the overall study contrast, if GPs in the control condition are aware of the study objectives, necessitating further clarification on what will be told on the RCT to the group of health care centres prior to randomization.

Randomization. To minimize the risk of imbalance, stratification will take place based on the ratio of patients to family doctors. While this may be sufficiently adequate, I wonder whether differences in 'culture' between professionals (related to both general practice and the secondary care), or other cultural aspects of the population itself (e.g. type of diet, behaviours, etc.) may also need to be taken into account (e.g. to avoid the risk of comparing the north to the south in the event randomization leads to such a division, which is not impossible, given the relatively low number of clusters).

Overview of Process and Design. In step 3 it is mentioned that the Ministry of Health will provide a list of 20 health care clinics will be provided to the working group. Which criteria will be used to yield this list? It might be useful to describe this in the manuscript, or appendix, along with an attempt to explore whether this selection sufficiently reflects a group of representative clinics.

Outcomes. From my own experience, it takes years to change routines in daily practice, so perhaps a follow-up of twelve months is relatively short. Therefore, it may be rational to expect small changes over this period only (in the introduction a relative reduction of 50% of hypertension-related hospital admissions is mentioned, which may impress as unrealistically high). If possible, it would be helpful to quantify such an expectation, to assess whether the suggested random samples will be sufficient to detect changes between clinics. A limitation of using health records for the outcome might be both selective reporting (confounding towards higher risk) and Hawthorne effects (study awareness may change reporting behaviour and lead to recorded readings for a broader group, e.g. with lower risk or with regression-to-the mean effects, which may inflate the expected improvements in the intervention group). Perhaps it might be possible to include an additional control group, e.g. a few control practices outside the study design, to study these potential confounding effects?

	Independence of methods used. It appears that the steering group provides support to the clinics, but is simultaneously involved in the study group: is sufficient independence guaranteed? For instance, if GPs receive a training course by the support group and are also evaluated by them, they may not feel free to talk open about its limitations, or their failures to incorporate new knowledge or skills in their daily practice. Statistical analysis. This section may benefit from further clarification. For instance, will a multi-level approach be used, using a random slope for study centre, to account for differences between centres? (Similarly: for GPs and nurses?) Also, since this aims to be a mixed methods study, it might be useful to perform a post-hoc analysis on effectiveness on BP outcomes for individual centres and to link these results to the qualitative findings from the qualitative substudy. N.B. Line 242: replace 'implementation' by 'randomization'? PM: Is the study registered in a trial registry?
--	---

VERSION 1 – AUTHOR RESPONSE

Reviewer 1 – Dr. T. N. Nonten Dear Dr. Nonten, Thank you for your thoughtful peer review. Please find below our response to each of the comments provided. Best, Dr. D. Collins	
Reviewer Comment	Response
“Statistics are now not clearly described, please add this.”	Thank you for your comment. While a detailed statistical analysis plan is beyond the scope of this manuscript, we have added additional information to the data analysis synopsis. It now reads: “The change in indicators from baseline to follow-up will be calculated for intervention clinics and compared with control clinics (Table 1). Subgroup analysis by age, gender, and other demographic features may be done as deemed appropriate by the national steering committee. All analyses will account for stratified sampling. Since the health centre is the unit of inference for the outcomes (e.g. health centre proportion of eligible patients with a documented CVD risk score), use of an intracluster correlation coefficient is not required for analyses of these outcomes. Age and gender adjusted logistic regression models will be used to analyse the

	differences in pre-defined indicators between intervention and control clinics and between baseline and follow-up. The differences in means of continuous variables between the intervention and control clinics and baseline and follow-up will be analysed using age and gender adjusted analysis of variance.” (Line 340)
“Methodology: randomization and intervention is not blinded. This might not be required, but then authors need to think about possible bias in the control group. For example: do control group practices also have access to protocols? They might also want to 'implement' something in line with the intervention arm (dilution of the effect). Please consider this.”	Thank you for your comment. We agree that this is a potential source of bias. Given that the intervention is a complex one, it is not possible to blind the participants, health staff, patients, or outcome assessors. We have considered this point and clarify that the control group practices do not have access to the protocols. This is shown in Figure 1. We will also ensure to consider these points in the interpretation and discussion of the findings.

Reviewer 2 – Dr. H. H. Abholz	
Dear Dr. Abholz,	
Thank you for your thoughtful peer review. Please find below our response to your comment.	
Best,	
Dr. D. Collins	
Reviewer Comment	Response
“I would not publish Research Applications or pure Descriptions of a planned study- and this is one”	Thank you for your comment. It is our understanding that in its commitment to improve the quality of health research, The BMJ Open considers publishing research protocols including descriptions of a planned study. Our intent in publishing is in the spirit of improving the quality of research conducted, to reduce research waste, to increase methodological capacity particularly in the space of cardiovascular risk assessment in low-resource settings, and to provide a transparent prospective record of the study design and methodology. A statement from the BMJ Open website reads: “Publishing study protocols enables researchers and funding bodies to stay up to date in their fields by providing exposure to research activity that may not otherwise be widely publicised. This can help prevent unnecessary duplication of work and will hopefully enable collaboration. Publishing protocols in full also makes available more information than is currently required by trial registries and increases transparency, making it easier for others (editors, reviewers and readers) to see and understand any deviations from the protocol that occur during the conduct of the study.”

Reviewer 3 – Dr. Martin Gulliford

Dear Dr. Gulliford,

Thank you for your thoughtful peer review. Please find below our response to each of the comments provided.

Best,

Dr. D. Collins

“The paper describes work that is important in its national context. It might be useful for countries in a similar situation. On the other hand, the work may not be particularly novel as many countries are addressing these issues. The activities comprise audit and quality improvement activities.”

Thank you for your comment. We agree that this work is important in its national context, and that it is useful for countries in a similar situation. While there are many initiatives planned or underway to improve cardiovascular disease prevention, to our knowledge this is the first description of adapting and piloting WHO essential NCD interventions for primary healthcare in low resource settings (WHO PEN), and can therefore be of use to an international community.

In addition to this, it is our understanding that in its commitment to improve the quality of health research, The BMJ Open considers publishing research protocols. Our intent in publishing is in the spirit of improving the quality of research conducted, to reduce research waste, to increase methodological capacity particularly in the space of cardiovascular risk assessment in low-resource settings, and to provide a transparent prospective record of the study design and methodology.

A statement from the BMJ Open website:
“Publishing study protocols enables researchers and funding bodies to stay up to date in their fields by providing exposure to research activity that may not otherwise be widely publicised. This can help prevent unnecessary duplication of work and will hopefully enable collaboration. Publishing protocols in full also makes available more information than is currently required by trial registries and increases transparency, making it easier for others (editors, reviewers and readers) to see and understand any deviations from the protocol that occur during the conduct of the study.”

I did not see dates during which the proposed work was to be conducted.

Thank you for your comment. We have added the following sentence to line 171: “The planned study dates are from

	September 2016 to May 2019.”
There are many abbreviations that are not initially explained.	Thank you for your comment. We have reviewed the abbreviations and corrected any that were not initially expanded at their first use in the main text of the manuscript. It is our understanding that in the Abstract, common abbreviations do not require expansion, however we understand that the BMJ Open editorial team will provide final formatting suggestions if required.
“The basis for randomisation is uncertain. Is it ethically justified to include a control group that will not benefit from these interventions? A step-wedge design might have been better in this context.”	Thank you for your comment. The protocol was reviewed and obtained approval from the Research Ethics Committee of the Nicolae Testemitanu State University of Medicine and Pharmacy of the Republic of Moldova. This is described in line 343 of the manuscript. The aim of the evaluation is to determine the feasibility of implementing and evaluating essential interventions for the prevention of cardiovascular disease in primary health care in Moldova, with a view toward national scale up. (Line 152) In a resource-constrained health system, we also question whether it is ethically justified to invest in interventions before having
	locally valid information on their implementability, acceptability, and simplicity. And not least without testing a whether routine data can be used to assess effectiveness and for quality assurance, monitoring, and evaluation should a national scale-up become a reality. (These points are reflected in the objectives outlined in lines 157 to 166). Ultimately a cluster design was chosen for these reasons, and approved by the research ethics committee as described above.

“From a scientific perspective, some aspects of the manuscript appear simplistic. For example, the analysis plan reads ‘The change in indicators from baseline to follow-up will be calculated for intervention clinics and compared with control clinics.’ Though it is not made clear what statistical procedures will be used nor how cluster randomisation will be accounted for.”	Thank you for your comment. While a detailed statistical analysis plan is beyond the scope of this manuscript, we have added additional information to the data analysis synopsis. It now reads: “The change in indicators from baseline to follow-up will be calculated for intervention clinics and compared with control clinics (Table 1). Subgroup analysis by age, gender, and other demographic features may be done as deemed appropriate by the national steering committee. All analyses will account for stratified sampling. Since the health centre is the unit of inference for the outcomes (e.g. health centre proportion of eligible patients with a documented CVD risk score), use of an intracluster correlation coefficient is not required for analyses of these outcomes. Age and gender adjusted logistic regression models will be used to analyse the differences in pre-defined indicators between intervention and control clinics and between baseline and follow-up. The differences in means of continuous variables between the intervention and control clinics and baseline and follow-up will be analysed using age and gender adjusted analysis of variance.” (Line 340)
“Overall, this plan describes good work, but it does not make a convincing case that the cause of science requires this protocol to be published.”	Thank you for your comment. It is our understanding that in its commitment to improve the quality of health research, The BMJ Open considers publishing research protocols, and that this is in a broader context to advance the cause of science and not least applied health research. Our intent in publishing is in the spirit of improving the quality of research conducted, to reduce research waste, to increase methodological capacity particularly in the space of cardiovascular risk assessment in low-resource settings, and to provide a transparent prospective record of the study design and methodology. A statement from the BMJ Open website: “Publishing study protocols enables researchers and funding bodies to stay up to date in their fields by providing exposure to research activity that may not otherwise be widely publicised. This can help prevent unnecessary duplication of work and will hopefully enable collaboration. Publishing protocols in full also makes available more information than is currently required by trial registries and increases transparency, making it easier for others (editors, reviewers and readers)

	to see and understand any deviations from the protocol that occur during the conduct of the study.”
--	---

Reviewer 4 – Dr. E. P. Moll van Charante Dear Dr. Moll van Charante, Thank you for your thoughtful and detailed peer review. Please find below our response to your comment. Best, Dr. D. Collins	
“In this article, the authors describe the protocol of a study aimed to determine the feasibility of implementing and evaluating essential interventions for the prevention of CVD in primary health care in Republic of Moldova. Population wide interventions on the prevention of NCDs, such as cardiovascular disease, in low- and middle income countries are both urgently needed and rare, and the authors (and other initiators involved) are to be commended on this initiative.”	Thank you for your positive comments, and recognition of the urgency and relevance of this work.

“Introduction -- Consider adding some information about the current treatment gap (see above) and a (realistic) window of opportunity. Also, the reader might want to understand how the University of British Columbia became involved in this project.”	Thank you for your comments. We have considered this and have added the following regarding a realistic window of opportunity: “Despite these health system challenges, the Government of Republic of Moldova is committed to improving primary health care capacity for NCDs. It is estimated that 60% of hypertension-related hospital admissions (about 12,000 annually) and 40% of diabetes-related hospitalizations (about 5,000 annually) could be prevented through strengthened primary health care for these conditions, including better identification and management of those at increased CVD risk.(11) Given the need and international policy support for addressing this gap in NCD care, there was a favourable window of opportunity to act with impact. As such, strengthening primary health care was set out as one of the main commitments in the Action Program of the Government of Republic of Moldova 2016–2018.(12) To do this requires the development of simplified clinical protocols, in-person training programs for nurses and doctors, and a core set of indicators to monitor and evaluate changes in the quality of care.” (line 120) At the project’s inception, D. Collins was a DPhil candidate affiliated with The University of Oxford and WHO Collaborating Centre for Self Care and NCDs, and is now affiliated with the University of British Columbia. We feel that describing this is beyond the purview of the study protocol introduction. However, given that it is described here, it will be a part of the open peer review record and available to all readers.
“Methods -- Study contrast. Next to new developments in the prevention of CVD, Hawthorne effects may reduce the overall study contrast, if GPs in the control condition are aware of the study objectives, necessitating further clarification on what will be told on the RCT to the group of health care centres prior to randomization.”	Thank you for your comments. We agree, the Hawthorn effect may add uncertainty to the effect estimates, and is an inherent source of bias in the study design. We will carefully consider this in the analysis and interpretation of the results. Baseline data were collected before randomization, at which time neither the clinics nor the data extractors knew the allocation. We have added the following to the manuscript:

	“This will be done by a specially trained group of postgraduate medical trainees, before randomization such that neither the
--	---

	clinics nor the data extractors will know the allocation of each clinic to intervention or control arm.” (Line 191) Thus all sites received that same information about the purpose of the project, but not specific details about the outcome measures. The intervention clinics will receive at least one inperson follow-up support visit (line 206 and Figure 1), and this will be considered in the interpretation of the results. This will include understanding the control as usual care plus two site visits to extract routine data. In a health system with electronic medical records, this source of bias in the control may have been limited as routine data could be collected without a site visit. However, in the context of Moldova, all data are captured in free text on paper records.
--	--

“Randomization. To minimize the risk of imbalance, stratification will take place based on the ratio of patients to family doctors. While this may be sufficiently adequate, I wonder whether differences in ‘culture’ between professionals (related to both general practice and the secondary care), or other cultural aspects of the population itself (e.g. type of diet, behaviours, etc.) may also need to be taken into account (e.g. to avoid the risk of comparing the north to the south in the event randomization leads to such a division, which is not impossible, given the relatively low number of clusters).”	Thank you for your comments. We agree that this method of stratification is sufficiently adequate. The steering group felt that stratification by ratio of patients to family doctors was sufficient, but your comments about differences in culture are well taken. It was felt by the steering group that differences in culture or aspects of the population were not large enough to justify stratification by other means. While this stage of the study is already complete, we will ensure to address these points in the analysis and interpretation of the findings.
---	---

“Overview of Process and Design. In step 3 it is mentioned that the Ministry of Health will provide a list of 20 health care clinics will be provided to the working group. Which criteria will be used to yield this list? It might be useful to describe this in the manuscript, or appendix, along with an attempt to explore whether this selection sufficiently reflects a group of representative clinics.”	Thank you for your comments. National policy dictates that the Ministry of Health must nominate potential study sites. We have expended the description in the manuscript as below: “Health facilities will be nominated by the Ministry of Health for participation based on the following eligibility criteria: (1) primary health care facilities must be operating in the public sector as legal entities; (2) primary health care facilities must be sampled in a way such that they are geographically distributed evenly across the country; equally from the Central, North and Southern regions of MDA; and (3) health facilities must be primary health care centres that are managed by family doctors with no specialist doctors working in the facility. These criteria were chosen in order to select a group of clinics that sufficiently reflect the majority of primary health care facilities in Moldova.” (Line 263)
“Outcomes. From my own experience, it takes years to change routines in daily practice, so perhaps a follow-up of twelve months is relatively short. Therefore, it may be rational to expect small changes over this period only (in the introduction a relative reduction of 50% of hypertension-related hospital admissions is mentioned, which may impress as unrealistically high). If possible, it would be helpful to quantify such an expectation, to assess whether the suggested random samples will be sufficient to detect changes between clinics. A limitation of using health records for the outcome might be both selective reporting (confounding towards higher risk) and Hawthorne effects (study awareness may change reporting behaviour and lead to recorded readings for a broader group, e.g. with lower risk or with regression-to-the mean effects, which may inflate the expected improvements in the intervention group). Perhaps it might be possible to include an additional control group, e.g. a few control practices outside the study design, to study these potential confounding effects? Independence of methods used. It appears that the steering	Thank you for your comments. We agree that 12-months is relatively short and that the random samples may not be sufficient to detect changes between clinics. However, the aim of the evaluation is to determine the feasibility of implementing and evaluating essential interventions for the prevention of CVD is primary health care, and the details are reflected in the objectives (line 158). Practically speaking, this was a time frame that was felt to meet the study aim while also being pragmatic with respect to resources. At this time we are unable to add additional study sites, but your comments about selective reporting and Hawthorne effects are well taken, and are an important aspect of the study design which will be considered in the analysis and interpretation of the results.

group provides support to the clinics, but is simultaneously involved in the study group: is sufficient independence guaranteed? For instance, if GPs receive a training course by the support group and are also evaluated by them, they may not feel free to talk open about its limitations, or their failures to incorporate new knowledge or skills in their daily practice.”

For the quantitative strand, the collection of baseline and follow-up data are performed by a specially trained group of postgraduate medical trainees (see line 186) and to clarify, these are a different group of people than those who conduct the training.

For the qualitative strand, data collection was done by four members of the national steering group. While some of these members were involved in follow-up support visits, allocation of primary health centers for interviews was done to ensure that the interviewers were paired with clinics with whom they had not provided support visits. This was done to reduce possible influence; however in keeping with qualitative analysis standards reflexivity will be employed throughout the analysis to consider how the researcher and their context affects the results. We have added the following to the Qualitative methods section:

“The interviews will be conducted by members of the steering group, but the interviewers will be allocated to participants from health centres with whom they did not provide follow-up support visits.”
(Line 346)

“Statistical analysis. This section may benefit from further clarification. For instance, will a multi-level approach be used, using a random slope for study centre, to account for differences between centres? (Similarly: for GPs and nurses?) Also, since this aims to be a mixed methods study, it might be useful to perform a post-hoc analysis on effectiveness on BP outcomes for individual centres and to link these results to the qualitative findings from the qualitative sub-study.”	Thank you for your comments. We have added additional information to the data analysis synopsis. It now reads: “The change in indicators from baseline to follow-up will be calculated for intervention clinics and compared with control clinics (Table 1). Subgroup analysis by age, gender, and other demographic features may be done as deemed appropriate by the national steering committee. All analyses will account for stratified sampling. Since the health centre is the unit of inference for the outcomes (e.g. health centre proportion of eligible patients with a documented CVD risk score), use of an intracluster correlation coefficient is not required for analyses of these outcomes. Age and gender adjusted logistic regression models will be used to analyse the differences in pre-defined indicators between intervention and control clinics and between baseline and follow-up. The differences in means of continuous variables between the intervention and control clinics and baseline and follow-up will be analysed using age and gender adjusted analysis of variance.” (Line 340) We agree that a post-hoc analysis on effectiveness on BP outcomes for individual centres linked with qualitative findings would add to the study. This is explained at a high level starting at line 225 of the manuscript. We have also added the following (line 336): “Integration of Quantitative and Qualitative Strands The resulting qualitative data will be analysed thematically using the framework approach, and used to help explain the findings of the quantitative strand.(14) Following the sequential mixed method design, integration of the qualitative findings with quantitative findings will allow for the interpretation of the results in light of each other. This may include post-hoc analysis of effectiveness analysis of some of the quantitative outcomes as appropriate, to further add meaning to the integration of qualitative and quantitative strands.”
“Line 242: replace ‘implementation’ by ‘randomization’?”	Thank you for your comment. We have made this change.

VERSION 2 – REVIEW

REVIEWER	T.N. Bonten Leiden University Medical Center Dept. Public Health & Primary Care Leiden, the Netherlands
REVIEW RETURNED	27-Mar-2019

GENERAL COMMENTS	The authors have adequately addressed my comments. I have no further questions or remarks.
--

REVIEWER	Dr. Eric P. Moll van Charante Dept. of General Practice, Amsterdam University Medical Center, the Netherlands
REVIEW RETURNED	22-Apr-2019

GENERAL COMMENTS	I feel that the authors have adequately addressed my main concerns and the suggested changes have improved the manuscript. There is only one remaining issue that –in my view- may just need a little further clarification, which is about what GPs and other health care professionals in the control centres know about the specifics of the ongoing intervention (potential Hawthorne effect). Are they well aware of the rationale of the study (large window of opportunity on cardiovascular prevention in the Republic of Moldova), and may they feel inclined to improve their quality of care on cardiovascular risk management (CVRM), that is currently given national attention? It might be helpful to address the potential Hawthorne effects by quantitatively exploring potential changes in registration of relevant aspects of CVRM and qualitatively studying GPs' efforts regarding their CVRM care, to optimally interpret the study results.
--

VERSION 2 – AUTHOR RESPONSE

Regarding Dr. Moll van Charante's (Reviewer 4) comment about what GPs and other health professionals know about the ongoing intervention, we have stated in the Protocol that the Control group do not get any intervention and that they do not receive training or the guidelines from which to change practice. Although unlikely, information might disseminate beyond our control in the real-life situation — but the training and practice materials cannot and these are integral to the intervention. We therefore have included in our protocol the quantitative measures suggested by Dr. Moll van Charante including registration of different aspects of CVRM - these are listed in Table 1 of indicators. The Protocol also includes a qualitative arm of GPS efforts in order to optimally interpret the study results, in keeping with Dr. Moll van Charante's comments. We feel these measures are sufficient to take into account the observer effect in the Protocol, but we will clearly address the limitations in Discussion of the evaluation when publishing the results. We thank Dr. Moll van Charante for emphasizing this important point.